# Influence of Epoxy Content on the Properties and Marine Bacterial Adhesion of Epoxy Modified Silicone Coatings

**Ruikang Zhao, Zhanping Zhang * and Yuhong Qi**

Department of Materials Science and Engineering Dalian Maritime University, Dalian 116026, China; zrk@dlmu.edu.cn (R.Z.); yuhong_qi@dlmu.edu.cn (Y.Q.)

**\*** Correspondence: zzp@dlmu.edu.cn; Tel.: +86-0411-8472-3556

**Abstract:** This study addresses the issue of enhancing the mechanical properties and adhesion of silicone antifouling coatings. In this paper, γ-aminopropyltriethoxysilane was used to pretreat bisphenol A epoxy resin to obtain epoxy-silicone prepolymer, which was then mixed with hydroxyl-terminated polydimethylsiloxane to obtain epoxy-modified silicone. It was cured with polyamide curing agent and dibutyltin dilaurate catalyst to form film, and a three-component epoxy-modified silicone coating was prepared. Fourier transform infrared (FTIR) spectroscopy was used to characterize its chemical structure. The effects of epoxy content on the surface properties, mechanical properties and antibacterial properties of the coatings were characterized by confocal laser scanning microscope (CLSM), contact angle measurements, tensile test and bacterial adhesion test. The results show that adding epoxy makes the adhesion of the coating at level 1 and the surface free energy of the coating was between 15–21 mJ/m². When its content is less than 22.1 wt %, the coating is in a ductile material state. Once it is higher than 22.1 wt %, the coating was in a brittle material state. As the content increases, material's hardness and fracture strength increases; elastic modulus decreases first and then increases, but bacteria removal rate decreases. The modification of the epoxy to silicone can effectively improve the adhesion and mechanical properties of the coating, while maintaining the characteristics of the low surface of the coating. It plays a positive role in improving the performance of silicone antifouling coatings.

**Keywords:** γ-aminopropyl triethoxysilane; hydroxyl terminated polydimethylsiloxane; epoxy; coating; bacterial removal

## 1. Introduction

Ships sailing in the ocean will inevitably be attached by a variety of marine organisms. This fouling will cause a large increase in the ship's own weight and fuel consumption [1]. In order to inhibit marine biofouling, painting with antifouling coatings is the most convenient and widely used method [2]. Silicone antifouling coating with low surface free energy is mainly composed of polysiloxane segments. The methyl group is linked to the Si–O main chain with a σ bond, which gives the polysiloxane segment a strong non-polarity, giving it the characteristics of hydrophobicity and low surface free energy [3]. This greatly reduces the adhesion of fouling organisms onto the coating surface, as it is a completely non-toxic coating, and achieves environmental protection and antifouling effects [4,5]. However, the mechanical properties and adhesion of the silicone antifouling coating to the substrate are poor [5,6], which is insufficient to meet actual use requirements. Epoxy resin has a high benzene ring density. The epoxy polymer with the curing agent has a cross-linked network structure on the molecular scale [7,8] so that it can obtain a lower water permeability coefficient, good mechanical

properties and good adhesion to the substrate [9–11]. Therefore, epoxy and silicone are complementary in terms of advantages and disadvantages. In summary, by using epoxy to modify the silicone resin offers the possibility of preparing excellent marine antifouling coatings. Some studies [12–15] also proved the possibility of combining the flexible segment structure of silicone resin with the cross-linked structure of epoxy resin. For example, Zheng et al. [16] successfully prepared epoxy modified silicone through a condensation reaction using silicone intermediate and epoxy resin and cured it at room temperature using a polyamide curing agent. A certain degree of phase separation was found when the content of epoxy was 44 wt %, but this had no effect on the mechanical properties of the coating. Polydimethylsiloxane is considered to be one of the best materials for modifying epoxy resins due to its excellent flexibility and thermal stability [12,13], and with the increase of the content of silicone, the modified product is prone to phase separation [16]. Therefore, most studies use epoxy as the main film-forming substance, and the content of silicone is relatively low. In view of this, epoxy modified silicone coatings were prepared with hydroxyl-terminated polydimethylsiloxane and bisphenol A epoxy as the two components of the film-forming substance in this paper. The content of the hydroxyl-terminated polydimethylsiloxane was from 35.7 wt % to 68.2 wt %. In order to improve their compatibility, γ-aminopropyltriethoxysilane was selected as the coupling agent. Pre-reaction of bisphenol A epoxy resin with γ-aminopropyltriethoxysilane, followed by mixing with hydroxy-terminated polydimethylsiloxane in proportion to prepare a film-forming component.

## 2. Materials and Methods

### 2.1. Materials

Hydroxy-terminated polydimethylsiloxane (PDMS) was obtained from Dayi Chemical Industry Co., Ltd., (Yantai, China). The kinematic viscosity of PDMS was 5000 mm$^2$/s. Bisphenol A epoxy resin (E51) was obtained from Nantong Xingchen Synthetic Materials Co., Ltd. (Nantong, China); epoxy equivalent is 184–195 g/mol. γ-aminopropyltriethoxysilane (LT-550) at a purity of 96 % was obtained from Hubei Bluesky New Material Inc. Dibutyltin dilaurate (DBTDL) was obtained from Tianjin Kemiou Chemical Reagent Co., Ltd., (Tianjin, China) as analytical grade. Polyamide (650) was produced by Sanmu group (Yixing City, China).

### 2.2. Preparation of Coating Samples

The coating is composed of three components. The component A is prepared with PDMS, E51 and LT-550 according to the formulation ratio. The component B is polyamide as a curing agent. The component C is DBTDL as a catalyst. Firstly, E51 and LT-550 were mixed to produce the ring-opening reaction; then, reacted product was mixed with PDMS and used BGD 750 Versatile Sand-Milling Dispersing-agitator (Guangzhou Biuged Laboratory Instrument Supplies Co., Ltd., Guangzhou, China) at 1500 rpm blending for 20 min. After blending was completed, the mixed liquid was poured into a tinplate container as component A of the coating system. Before painting, component A and component B were mixed according to the formula ratio, stirred well and cured for half an hour. By adding the component C and stirring evenly, the coating was finally prepared.

In order to study the interface performance and bacterial removal ability of the coating, the coating was brushed on glass slides with dimension of 75 mm × 25 mm × 1 mm. In order to study the adhesion and flexibility of the coating, the coating was brushed on tinplate with dimension of 120 mm × 50 mm × 0.3 mm. The finished coating was placed in a dust-free fume hood and cured at room temperature (25 °C) for 7 days. In order to study the mechanical properties of the epoxy modified silicone coating, poured into a Teflon mold with dimension of 150 mm × 150 mm × 2 mm for at least 7 days to form a cross-linked elastomer. We tried to make the coating thickness uniform throughout by placing it in a dust-free fume hood. The cast film sample was peeled off after being cured for 7 days and prepared with mold for test.

The formula of epoxy modified silicone coating is shown in Table 1. The "x" in ES-x represents the ratio of the mass of the epoxy to that of silicone and epoxy, multiplied by 100.

**Table 1.** Formulation of epoxy modified silicone coating (wt %).

| Sample | E51 | PDMS | LT-550 | Polyamide | Dibutyltin Dilaurate |
|--------|------|------|--------|-----------|----------------------|
| ES-10 | 7.6 | 68.2 | 17.0 | 3.8 | 3.4 |
| ES-20 | 14.9 | 59.7 | 14.9 | 7.5 | 3.0 |
| ES-30 | 22.1 | 51.5 | 12.9 | 11.0 | 2.5 |
| ES-40 | 28.9 | 43.5 | 10.9 | 14.5 | 2.2 |
| ES-50 | 35.7 | 35.7 | 8.8 | 17.9 | 1.9 |

## 2.3. Experiment and Characterization

### 2.3.1. Fourier Transform Infrared (FTIR) Spectroscopy

The molecular structure of the raw material and prepared coating were characterized by Frontier PerkinElmer infrared spectrometer (PerkinElmer Co., Ltd., Waltham, MA, USA) using the Attenuated Total Reflection (ATR) method. The scanning range is 4000–650 $cm^{-1}$. The resolution is 2 $cm^{-1}$. The number of scans is 32 times.

### 2.3.2. Contact Angle Measurement and Surface Free Energy Estimation

The contact angle measurement used deionized water and diiodomethane as the detection liquid. During the measurement, 3 µL of the detection liquid was dropped on the sample surface through the injection head provided by JC2000 contact angle measurement system (Shanghai Zhongchen Digital Technic Apparatus Co., Ltd., Shanghai, China). Three measurement points were selected on the surface of each sample, and the measured static contact angle was calculated and evaluated using JC2000D contact angle measurement software (version 1.0.0.1). The surface free energy of the sample was calculated according to the Owens two-liquid method [17], and the formulas are shown in Equations (1)–(3) $\theta_{H_2O}$ and $\theta_{CH_2I_2}$ respectively represent the contact angles of deionized water and diiodomethane on the sample surface. $\sigma_s^p$ represents the polar force. $\sigma_s^d$ represents the dispersion force. $\sigma_s$ represents the surface free energy of the sample.

$$\sigma_s^p = [(137.5 + 256.1 \times \cos\theta_{H_2O} - 118.6 \times \cos\theta_{CH_2I_2})/44.92]^2 \tag{1}$$

$$\sigma_s^d = [(139.9 + 181.4 \times \theta_{CH_2I_2} - 41.5 \times \cos\theta_{H_2O})/44.92]^2 \tag{2}$$

$$\sigma_s = \sigma_s^p + \sigma_s^d \tag{3}$$

### 2.3.3. Morphology Analysis

Morphologies of the coating surface and fractures were measured with the Olympus OLS4000 CLSM (OLYMPUS (China) Co., Ltd., Beijing, China) with a field of view 130 µm × 129 µm (surface micromorphology) and 646 µm × 645 µm (fracture morphology), using the scanning mode of XYZ fast scan. The line roughness ($R_a$) of the coating was measured using LEXT analysis software (version 2.2.4). The fracture morphology of the coating was analyzed using the samples obtained from the room temperature tensile test.

### 2.3.4. Mechanical Properties

BGD 560 Hinge-type Cylindrical Mandrel Tester (Biuged Laboratory Instruments (Guangzhou) Co., Ltd., Guangzhou, China) was used to test the flexibility of coating on the tinplate samples with a bend test according to national standard GB/T 1731 [18] (ISO 1519 [19]). The minimum shaft diameter (mm) that does not cause coating to crack was used to indicate its flexibility. BGD 503 Cross Cutting

Rule (Biuged Laboratory Instruments (Guangzhou) Co., Ltd., Guangzhou, China) was used to test the adhesion of the coating on the tinplate samples with a cross-cut test according to the national standard GB/T 9286 [20] (ISO 2409-2013 [21]). According to the coating thickness of the coating, selected the corresponding cutting distance to manually cut it with a single-blade cutting tool. After the adhesive tape was applied for 5 minutes, the adhesive tape was smoothly peeled off within 0.5–1 s. Finally, the test surfaces were classified according to the surface appearance of the cross zone cutting area. The QCJ film impactor (Tianjin Yonglida Material Testing Machine Co., Ltd., Tianjin, China) was used to test the impact resistance of the coating on the tinplate samples according to the national standard GB/T 1732 [22]. The maximum height (cm) of a weight of 1 kg falling on the sample without causing damage to the coating was used to indicate the impact strength of the coating. Bend test, cross-cut test and impact resistance test were performed on three samples of each formula. The TH220 Shore A hardness tester (Beijing Times Testing Instrument Co., Ltd., Beijing, China) was used to test the hardness of the coating (ISO7619 [23]). The cast film sample was folded several times until the thickness exceeded 5 mm. Six points were randomly take for testing and the average value taken as the result of the test. The UTM5105 Computer-controlled electronic universal testing machine (Jinan Wance Electrical Equipment Co., Ltd., Jinan, China) was used to perform tensile stress-strain tests on the cast film samples according to national standard GB/T 528-2009 [24] (ISO37-2005 [25]). It was cut into 45 mm × 4 mm dumbbell-shaped samples, and three duplicate samples were made for each sample. The test was performed at a tensile speed of 50 mm/min to obtain a stress-strain curve of the sample, and the elastic modulus, elongation and breaking strength of the sample were analyzed. Elastic modulus was fitted with the data that the strain was less than 0.02 mm/mm.

### 2.3.5. Bacterial Adhesion and Removal Test

The antifouling performance of the coating was characterized by bacterial adhesion and removal tests. The test bacteria were selected from the natural seawater of Yellow Sea in Dalian, China. The Samples of each formula were divided into two groups. Rinsing and washing were used to remove unattached bacteria and bacteria with lower adhesion to the surface of the coating, respectively. The bacterial removal ability of the coating was evaluated by comparing the bacterial removal rates obtained by two different methods of bacterial removal. Before the test, the measuring cylinders, beakers, cotton swabs, and sterile seawater were sterilized by YX-280D portable sterilizing equipment (Hefei Huatai Medical Equipment Co., Ltd., Hefei, China) under a high pressure of 0.1 MPa for 20 min. Then placed on the SW-CJ-1FD clean bench (Shanghai Boxun Industrial Co., Ltd., Shanghai, China) and sterilized by UV light for 20 min. Six coated glass slides (three for rinsing and three for washing) of each formula were immersed in natural seawater. Three coated slides of each formula were rinsed making up the group "Rinsed". Unattached bacteria on the surface were gently rinsed away with sterile seawater. The other three coated slides of each formula to be washed, making up the group "Washed", were put into a 50 mL centrifuge tube and filled with 40 mL of sterile seawater. HY-4 Variable-speed multi-fuction shaker (Guohua Electric Co., Ltd., Changzhou, China) was used to simulate seawater washing the surface of the sample for 15 min under the conditions of an oscillation amplitude of 20 mm and frequency of 130 r/min to removed bacteria with low adhesion to the surface of the coating. After washing and rinsing, each sample was put in 50mL of sterile seawater, using a cotton swab to brush the coating surface 6 times, and the bacteria attached to the surface of the coating was brushed into sterile seawater. Sterilized seawater was used to dilute the bacteria suspension to a volume of $10^{-6}$. A volume of 10 μL of the diluted bacterial suspension was inoculated on 2216E solid medium and spread evenly. The medium was then inverted and cultured in a biochemical incubator at 30 °C for two days. Pictures were taken to record the test results. Quantitative image processing was performed on the photos through Image-Pro Plus software. The picture was grayed out first, and then the contrast was enhanced. The processed bright areas of the image were counted; the average and standard deviation of the number of colonies on the surface of the three media of each group of samples were taken. The bacterial removal rate (*R*) was calculated from Equation (4). and represent the colony

concentrations of the rinsed and washed samples, respectively. Specific experimental methods and characterization of bacterial attachment refer to Huang et al. [26].

$$R = \frac{(C_{Rinse} - C_{Wash})}{C_{Rinse}} \times 100\% \tag{4}$$

## 3. Results

### 3.1. Molecular Structure

The infrared spectra of PDMS (liquid), E51 (liquid) and ES-30 epoxy modified silicone coating (solid) samples are shown in Figure 1. The symmetrical stretching vibration absorption peak of the epoxy group is at 912 cm$^{-1}$. Compared with E51, ES-30 has no absorption peak here, which indicates that the epoxy group has been completely consumed by the ring opening reaction. The benzene ring bending vibration absorption peaks of ES-30 are at 1610 cm$^{-1}$ and 1505 cm$^{-1}$, indicating that the crosslinked structure of epoxy resin has been successfully introduced into the siloxane structure. The formation process of the molecular structure can be explained as [27], the first stage is the pretreatment reaction between E51 and LT-550. When the two liquids with different viscosities are mixed uniformly, the epoxy group of E51 epoxy and the amino group of LT-550 undergo a ring-opening reaction. The principle of the reaction is shown in Figure 2.

**Figure 1.** FTIR spectra of polydimethylsiloxane (PDMS), E51 and coating ES-30.

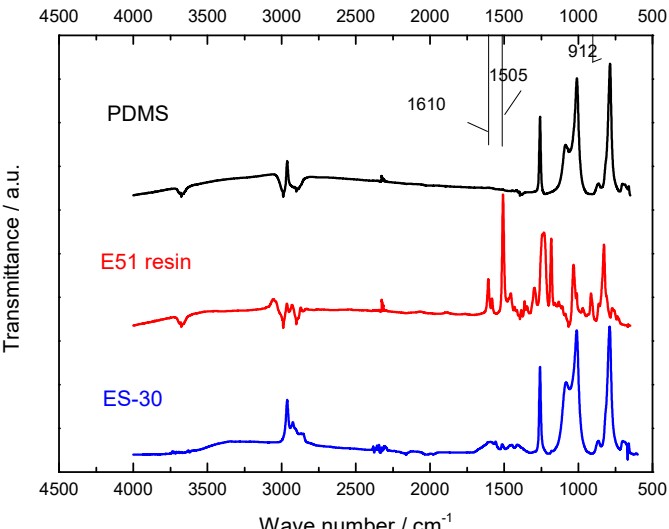

**Figure 2.** The reaction principle of epoxy group and LT-550.

The second stage is the condensation reaction between the hydroxyl group of PDMS and the ethoxy group of hydrolyzed LT-550 under the catalysis of DBTDL to complete the modification of silicone by epoxy [27]. It was found that the viscosity of the system increased during the mixing process. The reaction principle is shown in Figure 3.

**Figure 3.** The cross-linking reaction principle of epoxy modified silicone coating.

Finally, the polyamide curing agent is added to further cross-link the epoxy group that have not been fully reacted to increase the degree of cross-linking of the copolymer. The reaction principle is shown in Figure 4.

**Figure 4.** The curing reaction principle of epoxy resin and polyamide.

The five samples have similar characteristic peaks because their formulations are similar. As shown in Figure 5a, the characteristic peaks of the stretching vibration of the Si–Me$_2$ bond are at 790.5 cm$^{-1}$ and 886 cm$^{-1}$. The Si–O–Si bond symmetrical stretching vibration absorption peak is at 1010 cm$^{-1}$. The anti-symmetrical vibrational absorption peak of Si–O–Si bond is at 1068 cm$^{-1}$. By changing the addition ratio of epoxy and silicone, the characteristic peak intensity of the siloxane segment is changed. As shown in Figure 5b–c, as the proportion of epoxy resin increases, the characteristic peak intensity of the siloxane segment becomes weaker, and the characteristic peak of Si–O–Si also becomes weaker. The asymmetric stretching vibration absorption peak of –CH$_3$ is at 2964 cm$^{-1}$. No characteristic peak of epoxy group is found in the infrared spectrum, indicating that the epoxy group has completely reacted and the epoxy-modified silicone coating is successfully prepared.

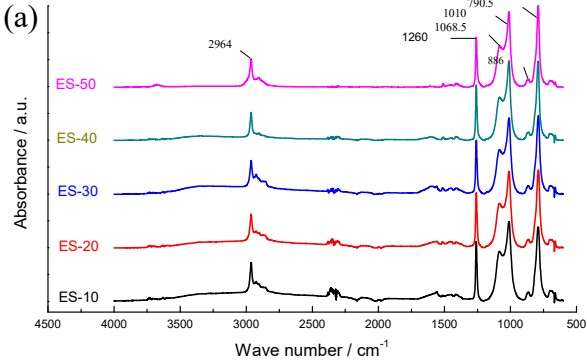

**Figure 5.** *Cont.*

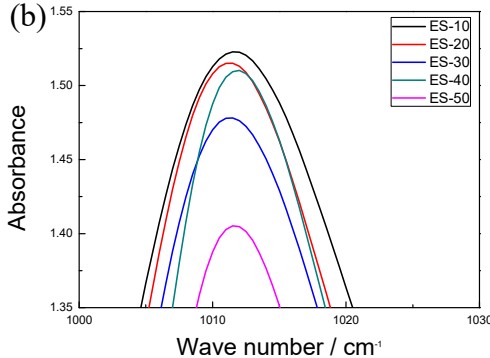
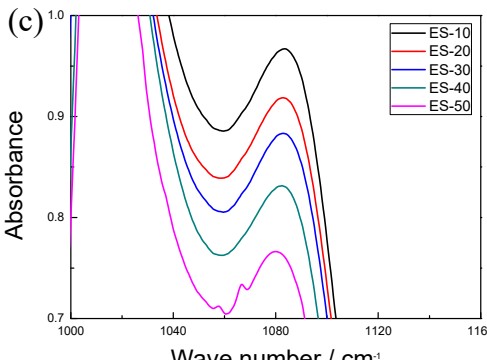

**Figure 5.** FTIR spectra of (**a**) epoxy modified silicone coatings; (**b**) the symmetrical stretching vibration absorption peak of Si–O–Si bond at 1010 cm$^{-1}$; (**c**) the anti-symmetrical vibrational absorption peak of Si–O–Si bond at 1068cm$^{-1}$.

### 3.2. Interface Performance

Interface performance is one of the important factors affecting the antifouling performance of epoxy-modified silicone coatings. As shown in Table 2, their water contact angles are measured between 105° and 115°, and the surface free energy is between 16 mJ/m$^2$ and 21 mJ/m$^2$, showing the characteristics of hydrophobicity and low surface free energy. This is due to the accumulation of silicone molecules on the coating surface during the curing process. The methyl group on the siloxane segment increases with the increase of the component of silicone, and the non-polar portion of the coating increases, thereby improving the hydrophobicity of the coating. The addition of epoxy resin cross-linked structure causes the micro-phase separation of the epoxy-modified silicone to increase, which makes the surface roughness of the coating increase with the increase of the epoxy content. When the roughness of the silicone rubber material is less than 6.63 μm, the water contact angle on the surface of the material will increase with the increase of the surface roughness [28]. Therefore, combining the non-polar part of the coating and the effect of its roughness on the hydrophobicity, epoxy-modified silicone coatings has similar characteristics of hydrophobicity and low surface free energy, and there is no obvious linear relationship. The surface free energy of the ES-30 coating samples is lower than other samples, indicating that the epoxy-modified silicone coating has excellent characteristics of low surface free energy at this ratio.

**Table 2.** Surface properties of the epoxy modified silicone coating.

| Sample | E51 Content (wt %) | Contact Angle (°) | | Surface Free Energy (mJ/m$^2$) | $R_a$ (μm) |
|---|---|---|---|---|---|
| | | Water | Diiodomethane | | |
| ES-10 | 7.6 | 113.67 ± 0.83 | 78.83 ± 1.53 | 19.68 ± 0.62 | 0.101 ± 0.072 |
| ES-20 | 14.9 | 109.06 ± 1.54 | 74.50 ± 1.17 | 20.67 ± 1.35 | 0.169 ± 0.059 |
| ES-30 | 22.1 | 114.25 ± 0.91 | 85.83 ± 1.03 | 16.45 ± 0.59 | 0.191 ± 0.044 |
| ES-40 | 28.9 | 113.17 ± 1.33 | 78.25 ± 0.47 | 18.15 ± 0.20 | 0.643 ± 0.067 |
| ES-50 | 35.7 | 112.19 ± 0.44 | 75.75 ± 1.17 | 18.87 ± 1.17 | 0.859 ± 0.095 |

### 3.3. Morphology and Microstructure

When the content of E51 is below 22.1 wt % (i.e., ES-10 to ES-30), as shown in Figure 6, each coating has good leveling property, and the surface is fairly smooth. With the increase of the epoxy content, the micro-phase separation of the coating surface is intensified, thereby increasing the roughness of the surface. Figure 7 shows the fracture morphology of the coating ES-50 by CLSM. Due to the poor compatibility between PDMS and E51 and the surface free energy of silicone resins being less than that of epoxy resins, delamination occurs after the paint is applied [29]. Therefore, it can be observed in the image that the fracture morphology of the upper area of the ES-50 tensile specimen (Figure 7a) has a relatively uneven fracture morphology compared to the fracture morphology of the lower area, which

deforms along the stretching direction to form the morphology of convex or concave (e.g., area marked with rectangle). This shows that the siloxane segments will agglomerate in the upper layer close to the coating surface, giving the low surface free energy. The upper area of the tensile fracture shows more ductile. The lower fracture morphology of the ES-50 tensile specimen (Figure 7b) is observed to have more smooth fracture morphology with a river-like texture (e.g., area marked with circle), which is caused by less deformation after brittle fracture. This shows that the crosslinked structure of the epoxy more easily gathers in the bottom area of the coating where fracture tends to more brittle.

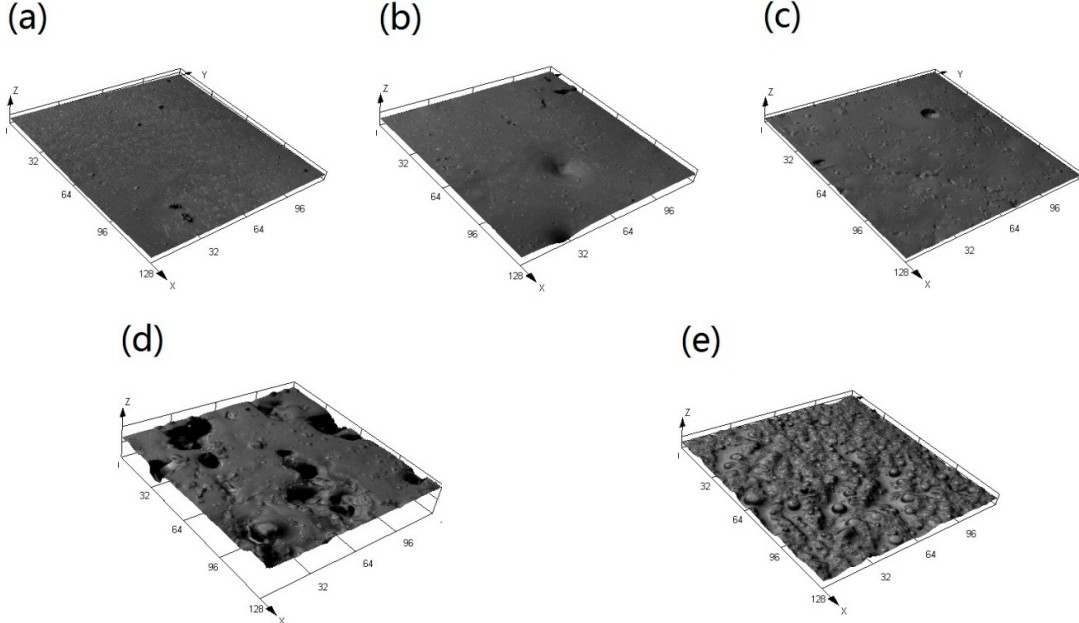

**Figure 6.** Morphology of the epoxy modified silicone coating: (**a**) ES-10; (**b**) ES-20; (**c**) ES-30; (**d**) ES-40; (**e**) ES-50.

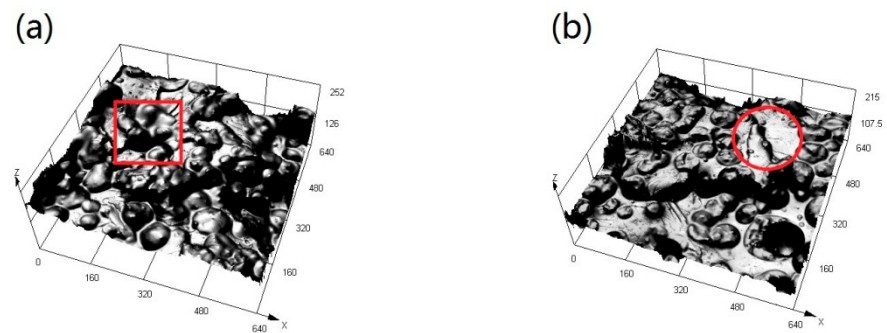

**Figure 7.** Fracture morphology of the ES-50 tensile specimen: (**a**) upper area and (**b**) lower area.

*3.4. Mechanical Properties*

3.4.1. Basic Mechanical properties

The basic mechanical properties of epoxy-modified silicone coatings are shown in Table 3. The coating has good impact resistance and flexibility, and the adhesion also reaches the level 1 (i.e., detachment of small flakes of the coating at the intersection of the cuts, but none of the squares are detached). As the epoxy resin content increases, the amount of the cross-linked network structure inside the coating becomes larger, and the hardness of the coating increases. Due to the addition of epoxy resin, the material state of the coating becomes brittle. When the content of E51 is 35.7 wt % (i.e., ES-50), Shore hardness of the coating can reach 89.40 HA.

**Table 3.** Mechanical properties of epoxy modified silicone coatings.

| Sample | E51 Content (wt %) | Shore Hardness (HA) | Adhesion (level) | Flexibility (mm) | Impact Resistance (kg·cm) |
|---|---|---|---|---|---|
| ES-10 | 7.6 | 5.90 ± 0.26 | 1 | 1 | 50 |
| ES-20 | 14.9 | 10.23 ± 0.25 | 1 | 1 | 50 |
| ES-30 | 22.1 | 13.97 ± 0.56 | 1 | 1 | 50 |
| ES-40 | 28.9 | 80.20 ± 3.16 | 1 | 1 | 50 |
| ES-50 | 35.7 | 89.40 ± 3.28 | 1 | 1 | 50 |

### 3.4.2. Tensile Properties

The elastic modulus, elongation and breaking strength of the coatings are shown in Table 4. Mean stress-strain curves of each coating are shown in Figure 8. When the content of E51 is from 7.6 wt % to 22.1 wt % (i.e., ES-10 to ES-30), the tensile curve of the coating appears to be soft and ductile [23]. As E51 content increases, the elastic modulus of the coating decreases first and then increases. When E51 content is less than 22.1 wt %, the addition of epoxy resin causes a decrease in elastic modulus due to the microphase separation inside the coatings. As E51 content continues to increase, the elastic modulus becomes larger. The coatings become less prone to elastic deformation. The introduction of an epoxy cross-linked network structure enhances the mechanical properties of the coating. As a result, the elongation of the coating becomes larger. When the content of E51 is 22.1 wt % (i.e., ES-30), its elongation reaches about 357 %. The breaking strength of the coating increases with the increase in E51 content. When the content of E51 is from 28.9 wt % to 35.7 wt % (i.e., ES-40 to ES-50), the tensile curve of the epoxy-modified silicone coating is stiff and brittle [30], which has higher elastic modulus and breaking strength, lower elongation. With the increase of the cross-linked network structure of epoxy resin, the coating's molecular mobility is limited, and brittle fracture occurs after the instantaneous elastic deformation during the stretching process, which appears as a brittle material.

**Table 4.** Tensile properties of coatings.

| Sample | E51 Content (wt %) | Elastic Modulus (MPa) | Fracture Elongation (%) | Fracture Strength (MPa) |
|---|---|---|---|---|
| ES-10 | 7.6 | 1.37 ± 0.05 | 130.45 ± 6.96 | 0.19 ± 0.02 |
| ES-20 | 14.9 | 1.30 ± 0.01 | 211.84 ± 3.88 | 0.19 ± 0.02 |
| ES-30 | 22.1 | 1.26 ± 0.01 | 357.23 ± 2.84 | 0.27 ± 0.01 |
| ES-40 | 28.9 | 2.02 ± 0.23 | 17.88 ± 0.24 | 0.24 ± 0.02 |
| ES-50 | 35.7 | 2.68 ± 0.22 | 38.20 ± 0.77 | 0.48 ± 0.04 |

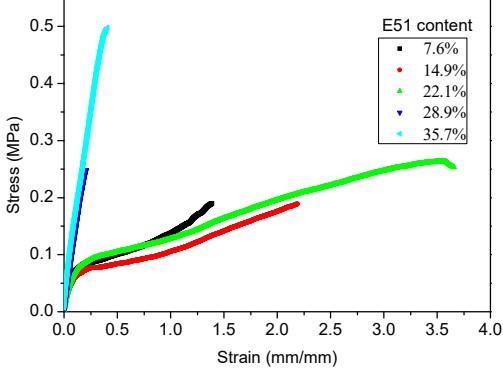

**Figure 8.** Mean stress-strain curve of epoxy modified silicone coatings.

## 3.5. Marine Bacteria Adhesion

Marine bacteria adhesion experiments are performed for coatings with different epoxy resin contents to evaluate the change in antifouling performance. Table 5 shows the original and processed images of marine bacterial colonies. It is observed that the washed samples had significantly less colonies than the rinsed samples, indicating that the epoxy-modified silicone coatings exhibit easy removal characteristics. Table 6 shows the results of the bacterial adhesion and removal test. According to the experimental results, it can be shown that with the increase of the epoxy resin content, the removal rate of bacteria on the coating surface decreases.

**Table 5.** Original and processed images of marine bacterial colonies.

| Sample | Original Image | | Processed Image | |
|---|---|---|---|---|
| | **Rinsed** | **Washed** | **Rinsed** | **Washed** |
| ES-10 | | | | |
| ES-20 | | | | |
| ES-30 | | | | |
| ES-40 | | | | |
| ES-50 | | | | |

**Table 6.** Bacterial adhesion test results and relative adhesion force between fouling organisms and coatings.

| Sample | E51 Content (wt %) | Colony Concentrationv (×$10^6$ CFU·mL$^{-1}$) | | Removal Rate (%) |
| --- | --- | --- | --- | --- |
| | | Rinsed | Washed | |
| ES-10 | 7.6 | 110.5 ± 9.4 | 50.9 ± 2.1 | 53.87 ± 2.02 |
| ES-20 | 14.9 | 125.3 ± 6.4 | 71.8 ± 3.5 | 42.69 ± 1.34 |
| ES-30 | 22.1 | 129.4 ± 4.1 | 78.4 ± 9.2 | 39.43 ± 2.69 |
| ES-40 | 28.9 | 93.2 ± 16.8 | 59.1 ± 8.4 | 36.59 ± 2.42 |
| ES-50 | 35.7 | 97.7 ± 17.5 | 64.1 ± 7.2 | 34.32 ± 4.38 |

According to the comparison of the surface roughness of the coating in Table 2 and the removal rate in Table 6, the factors affecting the ability of the coating to remove bacteria are studied. The bacterial removal rate of the coating decreases as the coating surface roughness increases. For the silicone coating, fouling organisms are more likely to adhere to coatings with rough surface morphology [31]. According to the microscopic morphology of the coating observed in Figure 7, it can be seen that the larger the epoxy resin content and the more irregular the surface of the coating, the lower its bacterial removal.

## 4. Discussion

### 4.1. Effect of Relative Bacterial Adhesion

According to the formula proposed by Brady [32], Ar represents the relative adhesion of the coating, E represents the elastic modulus of the coating, and γ represents the surface free energy of the coating. Ar is an important factor in characterizing low surface energy coatings. The relative adhesion value of the coating affects the strength of fouling organisms to adhere to its surface. As shown in Figure 9, as E51 content increases, the relative adhesion value of the coating decreases first and then increases, the bacteria removal rate decreases. In other words, there is no linear relationship between the bacterial removal rate of the epoxy-modified silicone coating and the relative adhesion value of the coating. This is due to the increase in the roughness of the coating, which reduces its surface energy, but the increase in roughness is an important factor in promoting bacterial adhesion. Therefore, the bacterial removal rate of the coating cannot be explained solely by the relative adhesion value of the coating surface.

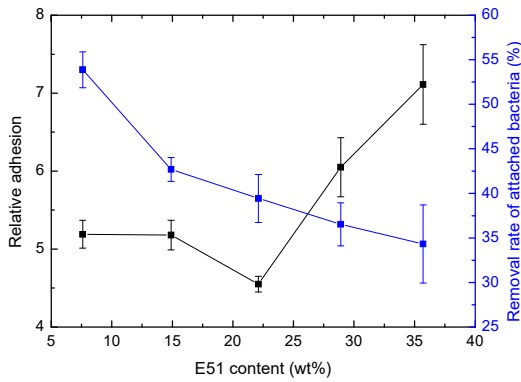

**Figure 9.** Relative adhesion and bacterial removal rate with E51 content.

### 4.2. Effect of Roughness on Bacterial Adhesion

As shown in Figure 10, with the content of E51 increases, the roughness $R_a$ increases, and the bacteria removal rate decreases. According to the attachment point theory proposed by Scardino [33], the microstructure of the coating surface can reduce the number of attachment points of fouling organisms, making it difficult to adhere to the coating surface, but the attachment point theory only

targets large-scale fouling organisms. However, for tiny creatures such as bacteria, the microstructure of the coating surface not only fails to inhibit bacteria from attaching to it but even provides more attachment points for bacteria, which will significantly increase its amount of attachment. According to the observation of CLSM, the micro roughness spacing of the coating is much larger than the size of bacteria. Not only does it not reduce the number of attachment points on the surface but it also provides more surface area onto which bacteria can attach, making attachment to the coating surface easier.

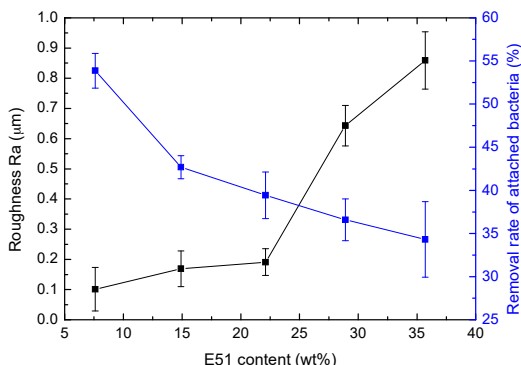

**Figure 10.** Roughness Ra and bacterial removal rate with E51 content.

### 4.3. Effect of Hardness on Bacterial Adhesion

As shown in Figure 11, with E51, content increases, hardness increases and, the bacteria removal rate decreases. Hardness is a measure of a material's ability to resist external deformation. For studied coatings, the smaller the Shore's hardness, the easier it is to produce elastic deformation, and the fouling organisms are more likely to peel off. Conversely, the greater the hardness of the material, the less likely it is to undergo elastic deformation, and the fouling organisms tend to fall off it in a shearing manner, needing more energy to peel from the surface of the coating.

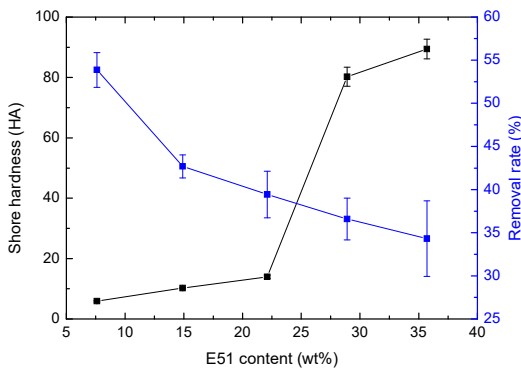

**Figure 11.** Shore's hardness and bacterial removal rate with E51 content.

## 5. Conclusions

In this study, epoxy E51 was used to modify PDMS, and epoxy-modified silicone coating was successfully prepared. The effect of the amount of E51 on the performance of epoxy-modified silicone coatings was discussed. The results showed that the surface free energy of the epoxy-modified silicone coatings with different epoxy resin contents was between 15–21 mJ/m$^2$, which showed the characteristics of low surface energy. When the content of epoxy resin was 22.1 wt %, the surface free energy of the coating was the lowest. The combination of the flexible structure of silicone resin and the cross-linked structure of epoxy resin changed the tensile properties of the coating, when the content of epoxy resin was 7.6–22.1 wt %, the coating was in a tough material state. When the content was 28.9–35.7 wt %, the coating was in a brittle material state. The addition of epoxy resin improved the mechanical properties of the coating and had excellent adhesion. As the content of epoxy increased,

coatings' hardness increased, fracture strength increased, elastic modulus decreased first and then increased, but bacteria removal rate decreased. Bacterial adhesion was affected not only by the elastic modulus and surface free energy but also by the roughness and hardness. With the increase of the roughness and hardness of the coating, marine bacteria adhered to it more easily, and the amount of attached bacteria increased. The higher the hardness of the coating was, the more difficult for the attached bacteria to fall off, which reduced the rate of bacteria removal. Therefore, the siloxane segments agglomerated on the surface of coating to provide the coating with good antifouling performance.

**Author Contributions:** R.Z., Z.Z., and Y.Q. designed and performed the experiments; R.Z. and Z.Z. analyzed the data; Z.Z. and Y.Q. contributed reagents/materials/analysis tools; R.Z. and Z.Z. wrote the paper. All authors have read and agreed to the published version of the manuscript.

**Funding:** This research was funded by National Natural Science Foundation of China (51879021), Project of Equipment Pre-research Field Fund (61409220304).

**Conflicts of Interest:** The authors declare no conflict of interest.

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
