# Peer review of "Influence of Epoxy Content on the Properties and Marine Bacterial Adhesion of Epoxy Modified Silicone Coatings"

_coatings, doi:10.3390/coatings10020126_

Round 1

Reviewer 1 Report

GENERAL OVERVIEW

The paper addresses the issue of enhancing the mechanical properties and adhesion of silicone antifouling coatings, by combining the beneficial properties of silicone and epoxy resin, which are complementary. An extensive experimental activity is designed to characterize 5 coating formulations, which differ from the epoxy content in silicone. The results are interesting and reasonably well-presented, therefore I recommend publication in "Coatings", prior some emendation to minor concerns, with particular reference to English style (which is fairly good anyway) and sentences arrangements. The paper is compliant with the journal’s scope.

ISSUES

The introduction is quite simple, but the background and the aim of the paper is concisely and clearly presented. However, in lines 49 - 51 "However, most studies had used epoxy as the main film-forming substance, and the content of silicone is relatively low.”, no references are provided to endorse this statement. References should be provided and discussed in depth.

Abstract: the abbreviations should be always disclosed: e.g. CLSM is never written in its extended form.

Verbal tenses should be checked (e.g. line 50: past perfect tense is used inappropriately; line 157 "epoxy modified silicone coating (solid) samples were shown in Figure 1": present tense is required)

Section 2.1 "Materials": properties of the materials that the Authors mention are declared by the manufacturers? It is advisable to provide one or more tables to gather the main properties of the raw materials.

Section 2.3.4 "Mechanical properties": for the sake of clarity, please provide more additional sketches/detailed information about test procedures, set-up and geometry of specimens. The only reference to guidelines is insufficient. 

Section 2.3.5 "Bacterial adhesion and removal tests": this section is confused. Information and test procedures should be rearranged in a clearer way to be better understood. Moreover, some sentences appear mistaken in terms of verbal tenses and general formulation. (e.g. (not exhaustive) Line 146: “[...]. And sterilized seawater...”; Line 147: [...]. "Take 10 uL from it and..."; Line 149: "[...]. Take pictures to record... Perform grayscale ..."). Moreover, what do group C and group P refer to?

Line 136: "protable" instead of "portable". Maybe a typo?

Lines 172-173: "It was found that the viscosity of the system increased during the mixing process.”. Is it a qualitative observation or some tests were performed to endorse such statement?

Section 3 "Results": Consistency in the labels is strongly advised. Labels of the mixtures given in Table 1 (ES-XX) should be maintained throughout the paper. Please add the labels in Table 2 and refer to such labels in the discussion of the results, instead of or at least together with the epoxy percentage (cf. line 217 "When the content of E51 is below 22.1 wt%, [...]", should be modified, for example, in this way: "When the content of E51 is below 22.1 wt% (i.e. ES-10 to ES-30), [...]".

Lines 222-226 "it can be observed in the image that the siloxane segments will agglomerate in the upper layer close to the coating surface, giving the surface low surface free energy. The upper area of the tensile fracture showed more ductile. The crosslinked structure of the epoxy more easily gathered in the bottom area of the coating where fracture tends to more brittle." How can you draw such conclusions about ductility of fracture from the images of Fig.7? Please clarify how the reader can observe that evidence from the images. 

Figure 6 (caption (b)): a "n" maybe eluded the spell-check.

Line 236: Certainly "Level 1" refers to specific guidelines. Could you please provide some additional information about that? For instance, providing the threshold values of level 1?

Figure 8: from the statistical standpoint, the mean curve is more rigorous instead of a typical curve.

Line 254: "hard": maybe the Authors mean "stiffer".

Line 265 and 266: from section 2.3.4 it is not clear to me which is the distinction between "rinsed" and "washed" samples.

Line 267 "Table 5". More likely the Authors refer to Table 6.

Line 274 "To investigate the factors that affect the ability of the coating to remove bacteria". The sentence should be revised. It seems mistaken.

Line 297 "with E51 content increases": please check this phrase.

Lines 301-302 "However, for tiny creatures such as bacteria, not only cannot they be inhibited from attaching, they can even increase significantly". Albeit I can "grasp" the meaning of the sentence, it is unclear and should be revised.

Figure 9 to 11: it is necessary to provide standard deviation bands for all measured data.

Line 314: "[...]. And need more energy". Please revise this sentence.

Section 5 "Conclusions". As conclusions are reported in the manuscript, a mere summary of the findings of the experimental activity is reported. That is reasonable, however, some considerations about the best solution obtained by the tests for practical applications should be developed in depth. Please enlarge the section with a broader discussion, also with reference to the works cited in the introduction, in the light of the Authors' findings.

REVIEWER’S FINAL OPINION: Suitable for publication, pending minor revision.

Author Response

Dear Reviewer:

Thank you for your comments concerning our manuscript entitled "Influence of Epoxy Content on the Properties and Marine Bacterial Adhesion of Epoxy Modified Silicone Coatings" (ID: coatings-699096). Those comments are all valuable and very helpful for revising and improving our paper, as well as the important guiding significance to our researches. We have studied comment carefully and have made corrections which we hope meet with approval. We used "Track Changes" function in Microsoft Word, so you can easily see all revisions. Major revisons in the paper and responses to your comments are in the attachment. Please see the attachment.

Reviewer 2 Report

The current manuscript, titled “Influence of Epoxy Content on the Properties and Marine Bacterial Adhesion of Epoxy Modified Silicone Coatings” presents a preparation of epoxy-silicone prepolymer coating to work as antifouling layer in Marine industry. Antifouling painting in engineering marine structure is imperative for long service life of the component. The manuscript is accepted should the following comments are addressed:

Language is a problem throughout the manuscript; It must be polished. Technical and scientific writing must be followed to help readers understanding the paper objectives and its findings.   Acronyms should not be used in Abstract; if used elsewhere, they have to be defined first. In the experimental part, all instruments and conditions of measurements must be defined. How did authors measure morphology? what type of measurement; SEM, TEM,...? Authors just mentioned Olympus OlS4000,…Provide the instrument used and conditions of measurements   What do you mean flexibility? is it compression test? or tensile test? All tests have to be mentioned by their technical names and conditions of measurements?  It is not clear how did you measure flexibility and adhesion force. Discussing the results is Ok but again, language is the major concern in this manuscript.

Author Response

Dear reviewer:

Thank you for your comments concerning our manuscript entitled "Inflence of Epoxy Content on the Properties and Marion Bacterial Adhesion of Epoxy Modified Silicone Coatings" (ID: coatings-699096). Those comments are all valuable and very helpful for revising and improving our paper, as well as the important guiding significance to our researches. We have studied comments carefully and have made corrections which we hope meet with approval. We used "Track Changes" function in Microsoft Word, so you can easily see all revisions. Major revisions in the paper and responses to your comments are in the attachment. Please see the attachment.
